# Ending Commercial Lion Farming in South Africa: A Gap Analysis Approach

**DOI:** 10.3390/ani11061717

**Published:** 2021-06-08

**Authors:** Jennah Green, Catherine Jakins, Louise de Waal, Neil D’Cruze

**Affiliations:** 1World Animal Protection 222 Gray’s Inn Rd., London WC1X 8HB, UK; JennahGreen@worldanimalprotection.org; 2Blood Lion NPC, P.O. Box 1554, Hermanus 7200, South Africa; info@bloodlions.org (C.J.); management@bloodlions.org (L.d.W.); 3Recanati-Kaplan Centre, Wildlife Conservation Research Unit, Department of Zoology, University of Oxford, Tubney House, Abingdon Road, Tubney, Abingdon OX13 5QL, UK

**Keywords:** African lion, *Panthera leo*, commercial breeding, wildlife farming, management, gap analysis

## Abstract

**Simple Summary:**

In South Africa, African lions (*Panthera leo*) are bred on farms for commercial purposes such as tourism, trophy hunting, and the international traditional medicine market. Despite its legal status, South Africa’s growing lion farming industry is a contentious issue. In 2020, a high-level panel was appointed to review the policies, legislation, and management of breeding, hunting, trade, and handling of four wildlife species, namely rhino, elephant, leopard, and lions. In May 2021, it was announced that the government will stop issuing permits to new entrants into this industry as well as the issuance of hunting permits and will start amending permit conditions to prohibit breeding and exclude tourism interactions with captive lions, effectively ending the lion farming industry. In order to follow this line of action, a comprehensive, well-managed plan will be required to ensure a responsible transition away from the current industry. Here, using a “gap analysis” management tool, we outline some of the key considerations necessary for a responsible, well-managed exit from the lion farming industry in South Africa. We compiled key background information about the current state of the industry and use this information to identify desired management states and specific steps that could facilitate a successful phase out of lion farming.

**Abstract:**

African lions (*Panthera leo*) are commercially farmed across South Africa for sport hunting, tourism, and the international bone trade, primarily in Southeast Asia. Despite its legal status, South Africa’s growing lion farming industry is a contentious issue. In 2020 a high-level panel was initiated to review the policies, legislation, and management regarding the breeding, hunting, trade, and handling of four wildlife species, including lions. In May 2021, it was announced that the government intends to amend existing permit conditions to prohibit lion breeding and tourism interactions with captive lions, as well as to stop issuing permits to new entrants into the industry, effectively ending lion farming. In order to follow this line of action, a comprehensive, well-managed plan will be necessary to execute a responsible exit from the industry as it currently stands. Using a “gap analysis” management tool, we aim to: (1) outline some of the key considerations regarding the current state of the lion farming industry in South Africa; and (2) propose specific action steps that could be taken within five key areas (regulation, animal welfare, health and safety, equitability, and conservation) to help inform a responsible transition away from this type of wildlife farming in the biodiversity economy. For our gap analysis, we conducted a semi-systematic literature search to compile key background information about the current state of the industry. This information was then used to identify corresponding desired management states, and steps that could facilitate a successful phase out of lion farming in South Africa. We hope our approach helps identify key considerations for a responsible transition and can help aid decisions during the management of this process.

## 1. Introduction

African lions (*Panthera leo*) are commercially farmed across South Africa. Over the last decade, an industry that began as a handful of small-scale captive breeding operations has grown exponentially to hold a current captive population of up to 8500 lions housed across 300–400 facilities [1,2], contributing an estimated R500 million (US$42 million) annually to the South African economy [3]. As stated in the Biodiversity Management Plan for African Lions, the primary purpose of these commercial facilities is to breed lions for financial profit [4]. This remit differs distinctly from zoological institutions (that may breed or keep captive lions for conservation), or animal rescue centres and sanctuaries (that house them for protection and rehabilitation purposes) [5].

Commercial lion farming is reported to have emerged in South Africa in response to increasing market demands for lion products [6]. Lions were initially bred in captivity to supply canned hunting operations in the 1990s [7,8]. However, since 2008 they have also been slaughtered for their bones that have been exported in increasing numbers to Southeast Asia for the traditional medicine industry [9]. A significant number of tourism-based industries also benefit from commercial captive lion breeding via non-consumptive purposes. For example, cubs and young adult lions are offered for interactive ecotourism and volunteerism experiences to paying tourists [1,6,8].

Figure 1 illustrates the full list of various known opportunities for commercial use, demonstrating that lions in South Africa can be maintained within one distinct sector (e.g., specifically bred and used solely for canned hunting), or may be traded between sectors at different stages of their development. Specifically, lion cubs can be bred at a single tourist facility where they are maintained for their entire lifespan, or alternatively purchased or rented from specialised breeders and returned once they have outlived their suitability for their tourism function [5]. The extent to which individual lions are traded between these different sectors is currently unclear [8]. However, information obtained through Promotion of Access to Information Act (PAIA) requests indicates that movement of lions between facilities and provinces occurs [10]. A recent survey of 117 captive lion facilities reported that 65% of facilities examined should be considered ‘multi-purpose’, with 79% having engaged with the hunting sector and 66% having sold skeletons to lion bone traders [6].

With regards to legislation, lion farming has been permissible in South Africa under a number of different regulations at a national and provincial level [12,13]. Internationally, commercial trade in lions, their body parts, and derivatives are governed by the Convention on International Trade in Endangered Species of Wild Fauna and Flora (CITES) under Appendix B. At the 2016 CITES Conference of the Parties, it was agreed that South Africa would be permitted to export lion bones, provided they are sourced from the captive-bred stock, within annual quota limits (established by the South African Department of Forestry, Fisheries, and the Environment (DFFE)), and reported to the CITES Secretariat annually [2,6]. An annual quota of 800 skeletons was established for 2017, which was temporarily increased to 1500 in 2018, but subsequently reduced back to 800 [14], due in part to international criticism. No export quotas were published for 2019 or 2020.

Despite its legal status, South Africa’s growing lion farming industry is a contentious issue that has raised animal welfare and conservation conversations among both the scientific community and in wider public discourse [1,12]. Although the impact that farming has on wild lion populations in African range states (currently listed as Vulnerable (IUCN, 2021)) is reported to be minimal [15], its contribution to the conservation of wild lion populations has been questioned [16,17,18] alongside whether lion farming may be stimulating consumer demand for tiger bone ((*Panthera tigris*) currently listed as Endangered (IUCN, 2021)) in Southeast Asian markets [9]. In addition, researchers have also raised concerns about the negative impact of lion farming on South Africa’s economy through international tourism reputational damage [3,5], and the potential risks of lion farming for animal and public health [19].

Following a decision by the South African High Court in 2019, which determined that the lion skeleton export quotas set in 2017 and 2018 were unlawful and constitutionally invalid [20,21], and potentially spurred by increasing recognition of growing contention towards the industry, the Minister of the DFFE initiated a high-level panel (HLP) to review the policies, legislation, and practises regarding the management, breeding, hunting, and handling of four wildlife species, namely rhino, elephant, leopard, and lions [22]. Following a consultation process, the panel reported their conclusions and recommendations to the Minister in December 2020. The HLP recommendations received Cabinet approval in April 2021 and in May the Minister of DFFE publicly announced her intent to adopt the majority of the HLP recommendations, including to halt and reverse the domestication of lions, an immediate halt on sale of captive lion derivatives, the hunting of captive bred lions and tourism interactions, i.e., effectively ending the commercial captive lion farming industry in South Africa. The recommendations will now go through Parliament with Cabinet approval, although the timeline for this process remains unclear.

Herein, using a “gap analysis” as a management tool (see methods for full description), we aim to: (1) outline some of the key considerations regarding the current state of the lion farming industry in South Africa; and (2) in light of available information propose specific action steps that could be taken within five key areas (regulation, animal welfare, health and safety, equitability, conservation) to help inform a responsible, sustainable, and just transition away from the lion farming industry in South Africa. Although it is not our intention to provide specific solutions or time frames for transition management, we hope that flagging key areas of consideration will help maximise the likelihood of a successful transition and minimise chances of unintended negative impacts. More broadly, we hope that this gap analysis can also serve as a useful case study to aid decision making relating to the commercial captive breeding of other wildlife species elsewhere.

## 2. Materials and Methods

A gap analysis is a management tool that can be used to compare the current state of an industry with a desired future state, in order to establish the gaps between the two states and identify an appropriate course of action [23]. Gap analyses were first described in the business management literature in the 1980s [24] and have since been adapted for application across a range of different industries [25,26,27]. They are primarily used in this context for comparison of current performance and desired performance of organisational success or employee output [23]. Although the term ‘gap analysis’ can also refer to a conservation evaluation technique that identifies areas in which selected elements of biodiversity are underrepresented [28], for the purpose of our study we use gap analysis in the context of industry management (not as a conservation evaluation technique).

For this gap analysis, a semi-systematic literature search was conducted to compile key background information about the current state of the lion farming industry in South Africa. A systematic review of the scientific literature was conducted using an academic database (Web of Science), searching articles for ‘*Panthera leo*’ and ‘African lion’, with the Boolean operator ‘AND’ and the additional terms ‘commercial farming’, ‘commercial breeding’, and ‘South Africa’. The literature returned was then used in a ‘snowball approach’ to identify additional relevant sources (i.e., articles were not returned by the search terms but were cited in papers that were). Some relevant non-scientific literature identified by authors was also considered, namely government reports pertaining to lion farming in South Africa [15,21,22,29], an inspection report from national animal welfare council authority [30], and two media articles [31,32]. These government reports and grey literature were identified through the same ‘snowball approach’ described for the academic literature search. We also searched the DFFE of South Africa website, using the search term ‘lion farming’, to identify any reports published between 2015–2020 pertaining to the commercial lion farming industry.

Current management issues identified during this stage of the process were organised into one of five key focus categories: (1) regulation (i.e., adherence to legislation and proper record keeping); (2) animal welfare (i.e., the physical and mental well-being of lions); (3) health and safety (i.e., illnesses, injuries, and biosecurity risks posed to people and lions) (4) equitability (i.e., the financial well-being of people currently dependent on this industry), and (5) conservation (i.e., the survival of wild lions and other species). A team of eight researchers used this information to identify both corresponding desired management states, and specific steps that could facilitate a successful phase out of lion farming in South Africa. The researchers have backgrounds in animal welfare, conservation, and criminal law, and employment experience including the academic, NGO, and communications sectors, with particular expertise focusing on the lion breeding industry in South Africa.

## 3. Results

The systematic search of academic literature provided 31 articles (see Appendix A). An additional 11 articles were identified through the snowball approach (see Appendix B) and a further 7 sources made up of government reports and media articles were identified as being of value to feature in discussions as part of the gap analysis. Following discussions informed by these sources, our gap analysis model summarises the ideal ‘desired states’ to work towards for effective management of an industry phase out, across five key areas. We identify a number of action steps to aid the process towards a regulated, transparent, and well monitored transition (see Table 1).

### 3.1. Regulation

In its current state, the lion farming industry is governed by a patchwork of contrasting legislation (pertaining to captive lion breeding, trading, hunting, and keeping) across multiple provincial and national authorities, with disparities that leave legal loopholes which create opportunity for harmful and fraudulent activity [33]. For example, lion euthanasia is prohibited in the North West province, but lions that are not sold for breeding or hunting can be translocated to the neighbouring Free State province where euthanasia is permitted [6]. Similarly, minimum release times for captive hunts, i.e., the release time between a captive bred lion set free onto hunting farm and the actual hunt, varies across provinces. For example, North West province has the shortest release time of 96 hours and as a result many lions are transported to this province for captive hunts [10].

There is also a paucity of publicly available information concerning the scope and scale of the captive lion industry, presumably because a complete national audit has never been undertaken [5]. The absence of transparent and centralised baseline information (such as studbooks, veterinary records, lion farm registers, employment records, and other financial data), and the issuing authority for permits existing at the provincial level with no overarching national level oversight or record keeping, impedes the ability of relevant authorities to manage the industry and ensure that it is compliant with existing legislation [33]. For example, it is more challenging for South Africa to comply with annual CITES quotas given that the total number of skeletons produced by each province is not being properly documented [10,20].

Consequently, to help reach its desired state, we surmised that the lion farming industry would benefit from being effectively and transparently monitored during (and after) a time bound phase out of the industry, and that properly enforced penalties for any infringements of the law would be required to act as an effective deterrent. In terms of specific action steps, we identified that the creation and communication of new cohesive legislation (at provincial and national level), the collection of publicly available baseline data, a national database of pre-existing permits pertaining to this industry and the provision of training and funds to enforcement agencies represent priority areas that require particular consideration. These suggestions are reflective of the recommendations outlined in the high-level panel review report [29].

### 3.2. Animal Welfare

Animal welfare concerns associated with the lion farming industry have been well documented over recent years. In particular, facility inspections conducted by South Africa’s National Council of Societies for the Prevention of Cruelty to Animals (NSPCA) found substandard conditions including inadequate hygiene, insufficient diet, and a lack of necessary provisions at nearly half of the 95 lion farms inspected [30]. Other reported concerns include “speed breeding” practises whereby breeders remove young cubs from lionesses before they are weaned, to force a premature return to oestrus for faster breeding cycles, which can take a toll on the physical and mental wellbeing of both lioness and cub [8]. In addition, concerns resulting from high inbreeding and other poorly managed breeding (which can result in reduced genetic variation, low reproductive performance, increased cub mortality, and reduced immune competence) have been raised [8].

To achieve their desired management state, as a result of this gap analysis we concluded that lion farms should be required to provide lions the highest welfare conditions possible, whilst prohibiting any breeding of new cubs, slaughter or hunting of lions, and direct human contact with lions for commercial purposes, during a time bound phase out of the industry. To achieve this, we identified the need for trained veterinary professionals to ensure the prevention of new captive born cubs via appropriate methods (e.g., contraception, sterilization, or separate sex housing), the registration of all lions at captive commercial facilities on a centralised national database (for example via a robust and informed microchipping programme), and the benefits of making all animal welfare protocols pertaining to animal husbandry, health, enrichment, and euthanasia publicly available during the phase out of the lion farming industry.

### 3.3. Health and Safety

The lion farming industry poses an on-going biosecurity risk via potential disease transfer to staff, visitors, and the wider public [19]. More than 60 pathogenic organisms have been identified in African lions, among which are several species that can be transmitted from lions to other species, including humans [19]. Lion farms are likely to pose risks of zoonotic pathogen transmission to the public because a key part of the industry is ecotourism, where tourists have direct contact with lions on a regular basis—in some cases without basic hygiene protocols (e.g., hand sanitizing) [19]. Lion farms also pose a safety risk to workers (e.g., farm workers, slaughterhouse staff, and taxidermists), visitors, and local communities. Specifically, captive bred lions have inflicted injuries and mortalities at lion farms offering interactive experiences with direct animal contact [31] and incidents have also been reported of lions escaping their enclosures and running stray in the local area [32].

To help transition away from this current situation to its desired state, we surmised that appropriate biosecurity, health and safety, and management protocols should be established for the lion farming industry and made publicly available during a time bound phase out. With regards to specific action steps, we identified that measures such as the provision of appropriate personal protective equipment (PPE), upkeep of enclosure maintenance, protocols pertaining to animal husbandry, and responses to any outbreaks of zoonotic disease would also help safeguard animals and people working within the industry (and the wider public) during this time. As a further precautionary measure, we also identified that any facilities remaining open to tourists during the phase out period should operate on an observation-only basis to minimize the risk of zoonotic disease transmission and predator attacks.

### 3.4. Equitability

Lion farming is often described as a substantial contributor to job creation in the South African economy that provides a valuable source of income for hundreds of South Africans (most of whom are thought to be concentrated in the North West and Free State provinces) [3,5,6]. However, there are concerns that in the long-term, the lion farming industry may have a “net negative” impact on the South African economy. For example, one study quantified reputational damage to South Africa from supporting captive predator breeding at $2.79 billion in Net Present Value Terms (NPV) over a ten-year period [5]. In addition, the use of volunteer programmes (that feed revenue and free labour into some lion farms) has been criticised for depriving the local labour force of employment opportunities [5].

To help transition away from this current situation to its desired state, we surmised that individuals who are economically reliant (directly and indirectly) on lion farming should successfully shift to sustainable alternative forms of income generation during a time bound phase out of the industry. To aid this transition we identified sustainable business transition plans (for lion farm owners) and sustainable alternative livelihood protocols (for lion farm employees) should be created, potentially with assistance from the governmental, private, and NGO sectors where required. Given the likely gradual nature of an industry phase out, relevant training and support could be provided (for example, through the Sector Education and Training Authority (SETA), a vocational skills training organisation in South Africa), to relevant employees while the current generation of captive bred lions remain housed at lion farms to ensure employment viability in alternative sectors.

### 3.5. Conservation

Currently, captive-bred (including farmed) lions do not play any role in conservation breeding or wild release programmes, due in part to habituation with people, as well as genetic unsuitability from inbreeding and crossbreeding risking the introduction of genetic pollution to wild populations [11]. Despite this fact, the conservation impacts of the lion farming industry can be overplayed in this regard (Moorhouse et al., submitted). For example, some volunteer tourists pay to participate in ‘husbandry’ at commercial facilities under the marketing pretext that they are contributing towards predator rehabilitation and future release back into the wild [5].

There is also concern that the lion farming industry could be contributing to pressure on wild big cat populations. Specifically, although direct links between legal trade in farmed lion parts and the targeted poaching of wild lion populations in South Africa (and other range states) has yet to be evidenced [12], there is reasonable concern the situation could arise [29,35]. Moreover, given that tiger bone wine is consumed amongst urban public in China and Vietnam [14], and that lion bone is variously used as a substitute [36], concerns that the lion farming industry in South Africa could negatively impact wild tiger populations through demand stimulation should be considered [14].

Consequently, to help reach its desired state, we concluded that lion farms should not engage in activities that have could potentially have a negative impact on wild lion populations in South Africa and elsewhere during a time bound phase out. To achieve this, we identified the need for management protocols and public messaging guidance (to support efforts that would prohibit the release of lions in to the wild, the sale of farmed live lions, and the sale/stockpile of their derivatives), and the benefit of increasing enforcement capacity during this time (in anticipation of a similar situation to the South African rhino horn trade, whereby increased restrictions were initially contested by market participants, resulting in illegal wildlife trade activity) [6,39].

## 4. Discussion

Although the South African lion farming industry has burgeoned in recent decades, during the last five years there have also been several key developments that are likely to have limited several major components of its revenue stream. Noteworthy (but not exhaustive) in this regard is the USA’s suspension on imports of captive-bred lion trophies in 2016 [6], Safari Club International’s adoption of a policy stance opposing the hunting of captive-bred lions [40], the Southern Africa Tourism Services Association’s categorization of tactile interactions with all infant wild animals, canned hunting, and breeding of lions as unacceptable (as per its animal interaction guidelines [34]), the South African High Court ruling in 2019 that determined the annual lion bone quota was unlawful and constitutionally invalid [21], the reduced tourist and trophy hunter numbers as a result of the COVID-19 global pandemic [41], and Vietnam’s ban on imports of all wildlife derivatives also following the COVID-19 pandemic [42].

The decision made by the Minister of DFFE as a result of the high-level panel recommendations represents a critical moment in the on-going discourse surrounding this controversial commercial enterprise. Following the recommendations to end commercial lion farming in South Africa, a comprehensive and well-managed plan will be required to facilitate a responsible exit from the industry. To that end, this gap analysis summarises some of the management issues, their corresponding ideal ‘desired states’, and identifies a number of action steps to aid the process towards a regulated, transparent and well-monitored time bound transition that mitigates potential unintended negative consequences for the lions and people currently operating within the industry.

Given that the current management issues identified through this gap analysis ranged across diverse and complex areas (i.e., regulation, animal welfare, conservation, health and safety, and equitability), it is logical to assume that a wide range of specialised stakeholders will be required to successfully develop and implement an effective and responsible transition away from lion farming in South Africa. Consequently, we recommend the creation of a robust, collaborative process with open forums for addressing issues through information sharing and consensus-based decision-making (e.g., see [43]). Many specialised frameworks have been developed to address such multi-faceted processes [44]. Stakeholder analysis can identify the perceptions and roles of different actors and identify underlying inter-stakeholder conflicts whereas scenario techniques can provide a powerful tool to explore potential trade-offs between different stakeholder views [44]. Pre-emptive resolution of potential conflict between stakeholders can reduce overall costs for stakeholders and governments during the management process [44]. We also note that, especially in light of competing values, such a process may also benefit from being led by a non-advocacy and non-litigious body to earn the trust and support of everyone representing a diversity of values [43].

Once these types of forums have been established, a logical next step in the process would be the early development of a change management roadmap as a method to help systematically plan, assign personal or institutional accountability for, and facilitate effective communication throughout, an effective time bound strategy for industrial change. In particular, it has been posited that these types of wildlife management plans can benefit from the application of logic models (a management tool that identifies a list of actions to be taken with specific achievable outputs as a tool for organising information in an if-then sequence (see [45])). Likewise, the use of graphical management plans could add value by providing a strategic framework (with clear vision, goals, objectives, actions, outcomes, and outputs for practical decision making [45] to identify specific pathways to industry transition). To help maximise chances of success, efforts to reduce consumer demand for lion products should be based on human behaviour change concepts that adhere to theories of best practice [46]. Although it is not our intention to provide specific solutions or time frames for transition management, we hope that flagging key areas of consideration will help to maximise the likelihood of a successful transition and minimise chances of unintended negative impacts.

It is important to note that lions are just one of the many wild animal species that are captive bred for commercial purposes in South Africa [47]. Moreover, trade of captive-sourced wildlife and wildlife ‘products’ has become increasingly more common in recent decades across most taxa [48]. Possible reasons for the increased trade of captive-sourced wildlife include perceived reliability, quality assurance, public perception regarding exploitation of wild animals, controls on wild harvest, or declining availability of wild animals [48]. However, concerns have been raised, given that some captive-bred wild species stocks may impact negatively on wild populations [47], involve poor animal welfare conditions [49], and/or biosecurity risks [50] that are deemed untenable.

In light of the potential negative impacts associated with wildlife farming, formal policies to end the commercial captive breeding for certain wildlife species have been adopted in a number of different countries in recent decades. Notable examples include the decision to stop mink fur farming in the UK [51], to shift away from sea turtle meat farming on Réunion Island [52], and to close bear farms across South Korea and Vietnam [53,54]. More recently, in China there have been reports by state-run media outlets that an estimated 19,000 wildlife farms may have been shut down around the country following COVID-19 related concerns [55]. Given the possibility of similar decision-making in the future, we draw attention to the potential for this type of approach to be applied to other scenarios where a transition away from the commercial captive breeding of wildlife species is being considered and alternative solutions adopted.

In some scenarios, wildlife farms have made a successful transition away from commercial captive breeding and have instead focused on providing lifetime care for injured and/or confiscated wildlife. For example, Kélonia: Observatory of Marine Turtles (formerly a facility where personnel collected between 5000 and 20,000 wild green turtle hatchlings annually and raised them to marketable size) was transformed into a sea turtle research, education and rescue centre as a result of funding provided by The European Union and Regional Council [52]. Similar alternative facilities for lions potentially already exist within South Africa (e.g., between 11% and 13% of facilities surveyed in 2018 considered one of their purposes to be that of a sanctuary or rehabilitation centre for lions [6]) and could provide a useful starting point from which to expand a managed transition away from commercial operations. However, the extent to which this is feasible and/or manageable is not currently clear and requires further investigation.

Despite the potential benefits that a gap analysis can bring to wildlife management planning decisions, we acknowledge that there are some limitations associated with our approach. In particular, this gap analysis was carried out by a relatively small number of researchers, with experience and expertise gained via employment limited to the academic, NGO, and communications sectors, and was not based on a full systematic review of the current literature pertaining to the lion farming industry in South Africa. As such, it is possible that this analysis did not capture all of the management issues that would require consideration as part of an effective and responsible phase out. Despite this, we believe that this gap analysis provides a comprehensive, timely, and useful starting point that can be used alongside the recommendations of the high-level panel report to guide transition planning to end the lion farming industry in South Africa.

## 5. Conclusions

Lion farming in South Africa currently takes place against the backdrop of a controversial multifaceted debate [14]. There is no doubt the transition away from commercial captive lion breeding in South Africa will present significant change management challenges and great caution will be required to avoid unintended negative consequences for both lions and people. However, we posit that these challenges are not insurmountable and untenable negative impacts are by no means inevitable. In this regard, we suggest that the application of a gap analysis could be an effective tool as part of a wider change management approach (not only for the lion industry in South Africa, but for other wildlife farming scenarios involving other species elsewhere also).

## Figures and Tables

**Figure 1 animals-11-01717-f001:**
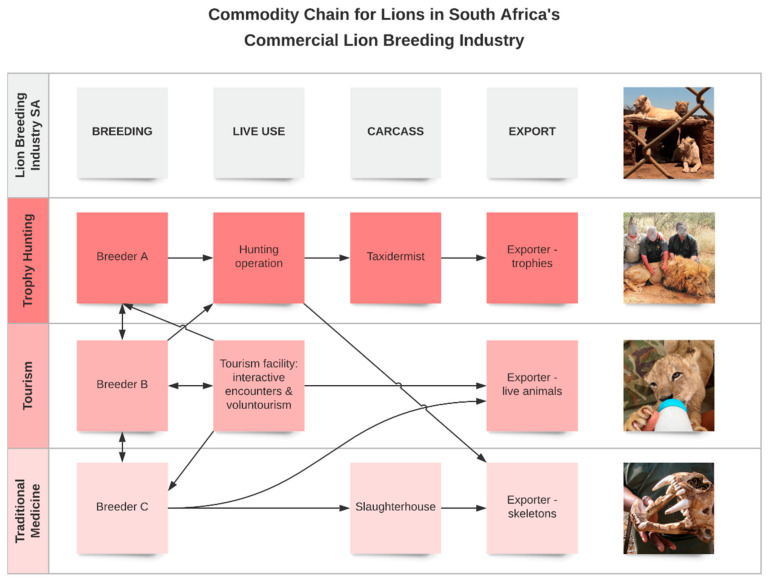
Commodity chain for lions in South Africa’s commercial captive breeding industry. The three recognised sectors currently involved in the commercial captive breeding of lions are the trophy hunting industry [8], non-consumptive lion tourism and volunteering [11], and international trade of lion bones to Southeast Asia for the traditional medicine industry [9]. This commodity chain highlights the potential for lions to move across sectors in relation to breeding, live use, slaughter and export throughout their lifespan, although the extent to which individual lions are traded between these different sectors is currently unclear [8].

**Table 1 animals-11-01717-t001:** Gap analysis table summarising five focus areas for consideration (regulation, animal welfare, health and safety, equitability, and conservation). Based on background literature and discussions with a team of eight researchers, the table also summarises the identified ‘current state’ (management issues associated with the present industry), the suggested ‘desired state’ (the perceived ideal management of an industry phase out) and proposed action steps (specific actions that could facilitate the transition between current state and desired state) for each focus area. See Appendix C for a more detailed summary of this gap analysis.

Focus Area	Current State (Commercial Lion Breeding Ongoing)	Desired State (Commercial Lion Breeding Ended)	Action Steps
Regulation	Commercial lion breeding in South Africa is currently governed through a patchwork of contrasting legislation (across a national, provincial, and departmental level) and penalties for infringements of existing legislation are not always properly enforced. Baseline data relating to the management of farms (e.g., lion numbers and studbooks) appear lacking. (Current state derived from a number of records from relevant government reports [15,21,22] as well as peer-reviewed publication detailing regulatory processes [5], and additional publications containing legal analyses regarding the governing of the lion farming industry [13,33]).	Commercial lion breeding facilities are effectively and transparently monitored during (and after) a time bound phase out of the industry. Penalties for infringements of the law are an effective deterrent and are properly enforced.	New cohesive legislation (at provincial and departmental level) is required to aid the phase out, in addition to resources and capacity provided for the relevant authorities to enforce new legislation. Baseline data detailing lion farm facilities are kept and made publicly available.
Animal Welfare	Current lion farming practises in South Africa are having a negative animal welfare impact for the lion’s physical and mental wellbeing. Additionally, the unregulated commercial captive breeding of lions in facilities is resulting in cases of genetic inbreeding. (Current state derived from consultation of an inspection report from the South African national animal welfare council authority [30], a court case brought forward to the High Court in South Africa by the National Council of the Society for Prevention of Cruelty to Animals [21] and a peer reviewed publication returned from our systematic literature search [8]).	Commercial lion breeding facilities provide lions the highest welfare conditions possible during a time bound phase out of the industry. Breeding of new cubs, slaughter, and direct contact with lions for commercial purposes are prohibited during this time.	The creation of veterinary protocols to prevent new lion cub births to underpin a phase out of the commercial captive breeding of lions, as well as the creation of animal welfare protocols (e.g., animal husbandry, health, enrichment, euthanasia) which are made publicly available, to ensure accountability.
Health and Safety	Commercial lion breeding poses a potential biosecurity risk via disease transfer to wildlife and people (staff, visitors, and wider public). Additionally, captive bred lions have inflicted injuries and mortalities at commercial facilities and have escaped from facilities, posing a potential biosecurity and safety risk. (Current state derived from consultation of one peer-reviewed publication detailing the risk of zoonisis transmission on the farms [19] as well as recommendations and risk considerations from the national Tourism Services Association [34] and media articles detailing misdemeanors with captive lions from farms [31,32]).	Commercial lion breeding facilities provide staff with the highest health and safety provisions possible during a time bound phase out of the industry. Any disease outbreaks originating from lions are effectively detected, contained, and eradicated during this time.	Biosecurity, health and safety and baseline management protocols relating to a phase out of the commercial lion breeding in RSA are created and made publicly available.
Equitability	The commercial lion breeding industry is currently owned by approx. 400 people across all known farms, including breeding, keeping, tourism, and hunting facilities. The industry employs approximately 1162 people across 4 provinces. (Current state of equitaibility derived from consultation of several peer-reviewed publications detailing the distribution of finances and employment opportunities within the sector [3,5,6]).	People dependent (directly and indirectly) on commercial lion breeding as an important economic source of income successfully shift to alternative sustainable livelihoods during a time bound phase out of the industry.	The creation of sustainable business transition plans, sustainable alternative livelihood protocols, and sustainable international donor fundraising plans (relating to a phase out of the commercial lion breeding in RSA), where required.
Conservation	Commercial lion breeding results in animals that cannot play an active role in wild release or conservation breeding programmes (e.g., due to inbreeding and habituation). Additionally, commercial lion breeding results in lion body parts and derivatives that are stockpiled and/or sold to meet demand for traditional Asian medicine (e.g., the biggest markets in China and Vietnam). (Current state derived from consultation of a number of peer-reviewed publications describing the impact of farms on wild lion populations and the relationship between the industry and the demand for lion bone exports for traditional Asian medicine markets [5,11,11,12,14,17,35,36,37,38]).	Commercial lion breeding facilities do not engage in activities that have a potential negative impact on the conservation of wild lion populations during (and after) a time bound phase out of the industry. The wild release, sale of live lions or their derivatives is prohibited during this time.	Conservation protocols (i.e., that differentiate between breeding for conservation versus commercial purposes) are created and made publicly available, in addition to management protocols for the disposal of lion body parts and other derivatives, and clear public messaging guidance relating to a phase out of the industry.

## Data Availability

Not applicable.

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
