# Peer review of "Ending Commercial Lion Farming in South Africa: A Gap Analysis Approach"

_animals, 2021, doi:10.3390/ani11061717_

Round 1
Reviewer 1 Report
As this article has the potential to inform ongoing policy discussions, it is important that literature from lion farming industry stakeholders is included. A ‘non-systematic literature search’ was conducted, but that stakeholder perspective can still be acknowledged in the framing of the study. The economies of the lion farming industry, for example, are not mentioned; nor are the potential economic implications of ending these practices.
Full comments below:
(20) ‘for necessary’ should be ‘necessary for’
(21) ‘in South.’ missing Africa
(35) ‘five key areas (regulation, animal welfare, conservation, equitability, and conservation)’ – four key areas as conservation has been repeated.
(73) ‘A recent survey of captive lion facilities’ – how many?
(99) Are these being farmed in the same facilities as lions?
(106) ‘Minister of the DEFF initiated a high-level panel’ – who is on this panel?
(117) ‘five key areas (regulation, animal welfare, conservation, equitability, and conservation)’ – four.
(133) ‘a non-systematic literature search was conducted’ – there should still be mention of literature from lion farming industry stakeholders within this study.
(136) ‘reports from industry stakeholders’ – what about literature from a lion farming perspective?
(138) ‘five key focus categories’ – there are only four.
(139) (3) is missing.
(140) ‘A team of eight researchers’ – only four are listed as authors of this paper. If not authors, how were these people recruited?
(148) ‘Health and wellbeing’ seems to be the missing area – see previous comments.
(246) ‘should successfully shift to sustainable alternative forms of income generation’ - such as? How?
(305) ‘public safety, and equitability’ – listed a health and safety previously.
(311) ‘led by led by’ – repeated words.
(324). ‘will to help maximise’ – help to.
(343) ‘in future’ – missing ‘the’.
(353) Has there been an example of a lion farm (or something similar) which has successfully shifted to a sustainable alternative?
(392) House style for references – media sources are incomplete.
Author Response
Please see attached word document.

Reviewer 2 Report
This is an interesting and well-written manuscript, and I have only minor suggestions.
The Introduction is very well-written, but maybe spoils some of the points treated in the Results. I would suggest trying to limit this effect by re-wording parts of the Introduction and leaving more unresolved some questions. In fact. the first objective of the manuscript seems already solved at a first reading.
I would suggest shortly defining gap analysis at the very first mention (i.e. move the first sentence of Materials and Methods - or a shorter version - to the last paragraph of the Introduction).
My main concern about this manuscript is the non-systematic literature review, which produces in the reader the feeling of an abritary choice by the Authors in selecting the literature to consider here. This feeling can be a major issue regarding manuscript reliability. I would suggest implementing a systematic literature review (I am sure the Authors will finally find the same articles), making explicit the search strings and the results (i.e. number of articles found and considered per topic, including a full article list in the Appendix).
Figure 2 is too large and partially redundant to the text of Results. I would suggest provide a more outlined version of this figure.
Author Response
Please see attached word document.

Reviewer 3 Report
My general comments are that this is a well-written article on an important issue, however the more I read the article, as it presently stands, the less convinced I was of it being a novel contribution to the literature and to this topic. I’ve noted the issues throughout, but generally I think there is a lack of specificity that hampers the worth of the information and argument being presented. For example, although I know very little about this issue, I also could probably have come up with “a need for standardized biosecurity protocols”- so the real worth of this method, the article in general, and the contributions of this author team should be in the process and the specifics of this statement, e.g. “X documents on zoonotic disease transmission incidents were consulted, and from the literature we argue that a neutral governing body with X numbers of veterinarians should be established, with such protocols as mandatory quarantine chambers, etc”. Having these kinds of specifics will make the article truly revelatory and truly a jumping off point for additional efforts.
Line 13: Clarify whether the “traditional medicine” lions are used for is in South Africa and/or Asia. My understanding is that the highest level of demand is in Asia, which entails international strategies and cooperation, versus traditional medicine use in South Africa which could be comparatively “easier” to tackle.
Line 20: “…considerations necessary for…”
Line 24: “should the decision to end…” this whole statement sounds a bit odd, I suggest rephrasing.
Line 27: Similarly to my earlier comment, clarify here, e.g. “international bone trade for traditional medicine, primarily in Asia”
Introduction
A missing component of the introduction is greater clarification of your aims- while I think the introduction is really well-written and does a good job of balancing the contentious parts of this issue, I think you need to strengthen the reader’s understanding that this gap analysis is for phasing out lion farming, and not a “neutral” analysis of what would be the best solution. I realized this because I was honestly surprised to get to Figure 2 (which is actually a Table), where the desired state is to have commercial lion breeding ended. In that respect, while again I appreciate the balance in the introduction, I think it would be better to “lean in” and (fairly) emphasize the negative aspects of lion farming and why it is at the point that it may be fully banned in 2021.
Line 59: Not just Southeast Asia- also China, e.g. Coals et al. (2020).
Line 119: Since you discuss many different aspects of lion trade/breeding, I think you should simply say “lion farming”, rather than constraining your comments solely to wildlife trade.
Lines 123 – 133: I had to come back to this section because I thought I missed something- I think there is a really significant gap here with how exactly you identified the states and steps in Figure 2 (which I think is really Table 1). I suggest looking at every box on the table under “Current State” and directly describing (briefly) how you “discovered” that in your methods. So for 1.2, writing something like “in our literature search we consulted X records from the South African Judiciary and found that penalties for improper lion breeding were inconsistent; for example in X case in X the penalty was $5 whereas in X case in X the penalty was $1000”. I think it is possible to summarize for certain aspects, but in general the methods needs a lot more clarity around how you got to the point of a fully fleshed out table.
Line 144 – 145: While I understand why this is relevant, I think this opens the authors up to criticism, considering how contentious this issue is, and how focused on potential ‘biases’ some researchers are. Ideally, I suggest thinking through how to rephrase and re-justify your non-systematic review- one example that comes to mind is Cerri et al. (2021), which also used a non-systematic review (called there a “scoping review”) and justified it according to e.g. the literature selection process. If for some reason this isn’t appropriate to your context and you wish to keep the focus on the researchers, then I suggest strengthening the argument, e.g. “authors on this team have given direct advice to South African Government Authorities based on 10+ years of experience collaborating with lion farmers”.
Cerri, J., Davis, E.O., Veríssimo, D. and Glikman, J.A., 2021. Specialized questioning techniques and their use in conservation: A review of available tools, with a focus on methodological advances. Biological Conservation, 257, p.109089.
Figure 2: General comments- there is some confusion with the parentheses, so I suggest going back over the table carefully and checking these. Also, while I understand the table is a summary of what you expand on further, I still would suggest including detail and providing more specific action steps beyond “X are created and made publicly available”… what should X be, specifically? How would they be publicly available? Who would be the intended audience?
Several comments I’ll address below:
- 2 and 2.3 are the same in “Action Steps”… how do they differ? What is mental domain versus physical domain? You explain in the “Current State” but it would be better to give more specific action steps connected with these different domains.
- 2: I recommend qualifying this to “e.g. the biggest markets of China and Vietnam”
- 3: what do you mean by “greenwashing”? I suggest removing greenwashing and instead being more specific here
Line 163: What types of harmful and fraudulent activity?
Line 171: “existing”
Line 184: This makes me think that there should be greater clarification of the aims of this paper, up in the introduction, as I’m not sure what the ultimate goal of this paper is. Defining these action steps is all well and good, but for greater impact I would also suggest sketching out, briefly, what would need to happen for these action steps to be accomplished- in which case you should also emphasize this in your introduction, too.
Line 199: Again, how will this be possible/achieved? How can farms be made to have the “highest welfare standards”, and how would that be measured? Etc
Line 270: How so, exactly? This should be explained in more detail.
Line 279: End parantheses.
Line 307 – 312: I think this all needs to be woven in directly and specifically with the proposed action steps detailed in the “Results” section, and clarified in the introduction that an aim is to identify action steps and provide management recommendations.
Line 318: Remove extra “that”
Line 323 – 324: Again, I think this definitely needs to be added into/clarified in the introduction.
Line 328 – 335: I get what you’re saying here but this first sentence is very confusingly worded and will definitely open you up to criticism, as to my knowledge there haven’t been studies quantifying type of trade (and if there are, they will definitely be heavily debated/criticized). Reword and clarify this entire section to draw your language back into stating that that wild trade still goes on, farmed trade has become more common compared to 20 years ago, and that farmed trade persists due to the reasons you describe- but not that farmed trade has necessarily become more common over wild trade, given the multiplicity of factors involved in wildlife trade.
Line 340: Also Vietnam (Crudge et al., 2018).
Crudge, B., Nguyen, T. and Cao, T.T., 2020. The challenges and conservation implications of bear bile farming in Viet Nam. Oryx, 54(2), pp.252-259.
Line 350: I think you are missing “where” here
Line 361 – 362: If you make this clear from the outset in your introduction, and throughout, then I think you can remove this limitation.
Line 353 – 366: This is an important point and one that I do think at present you don’t resolve. I am not convinced this article could be a “jumping off” point because another article that more thoroughly addressed the current literature would include the information you have here. Instead, I think you need to more directly and concretely explain your intentions, and the worth of this effort, in the ways I have described above, generally by being more explicit.
Line 375 – 377: Tighten up the wording here, there are some errors… Also, you don’t directly discuss what the problems could be. This should all go in the introduction- and would better frame your argument that you intended to look at what would be needed to end the practice.
Author Response
Please see attached word document.

Round 2
Reviewer 3 Report
While I recognize and respect the authors points that there is not enough information at present to make detailed recommendations on the level I Initially suggested, I think there are still some opportunities to present better conservation recommendations, based on other concrete and robust literature. I have noted these where I have gone through. Addressing these concerns I think can help to alleviate my continued thought that while this article is interesting and relevant, it could be rendered obsolete by a more concerted effort that does attempt to make detailed recommendations. In general, I see missed opportunities to make “brave” speculations, which could distinguish this article from following efforts. More simply- based on the evidence you do have, what could you theorize as concrete future conservation directions?
Table 1: While I recognize this is a summary table, I think there should still be appropriate references here, otherwise some of this can come across as anecdotal evidence (e.g. “Current lion farming practises in South Africa are having a negative animal welfare impact for the lions [<- additional note: missing the apostrophe here] physical and mental wellbeing”). I don’t doubt this is the case, but I think the evidence needs to be shown.
Line 285: Better explanation of SETA needed here for people unfamiliar with South African government bodies.
Line 301 – 302: What is meant here by “the preferred product”? I think it’s better to say something safer such as “consumed by the urban public”. As it reads now, it sounds like you are implying that tiger bone wine is the number one consumed wildlife product by individuals in China and Vietnam, which is almost certainly not the case.
Line 304: Impact how? (I suggest referring to Coals et al. (2020) for this).
Lines 336 – 345: I suggest consulting some of the human wildlife coexistence literature to give suggestions of concrete strategies for bringing stakeholders together and analyzing the most appropriate approaches. I know this is something HWC researchers have locked down- for example, check out the work of Alexandra Mejic, Jenny Anne Glikman, Jerry Vaske, and etc. I think including more specific information is important in light of the somewhat general nature of this article.
Line 346 – 359: And here, I suggest bringing in some of the theories found in the behavior change literature, as well as more general discussion of counterfactuals (e.g. a recent article by Coetzee and Gaston, 2021).
Author Response
Please see attached response.
